# Systematic Review: Neurodevelopmental Benefits of Active/Passive School Exposure to Green and/or Blue Spaces in Children and Adolescents

**DOI:** 10.3390/ijerph20053958

**Published:** 2023-02-23

**Authors:** Francisco Díaz-Martínez, Miguel F. Sánchez-Sauco, Laura T. Cabrera-Rivera, Carlos Ojeda Sánchez, Maria D. Hidalgo-Albadalejo, Luz Claudio, Juan A. Ortega-García

**Affiliations:** 1Paediatric Environmental Health Specialty Unit, Department of Pediatrics, Clinical University Hospital Virgen of Arrixaca, University of Murcia, 30120 Murcia, Spain; 2Environment and Human Health Lab, Instituto Murciano de Investigación Sanitaria (IMIB), University of Murcia, 30120 Murcia, Spain; 3Global Alliance to Renaturalize Child and Adolescent Health (GreenRooting.org), Spanish Association of Pediatrics, 28009 Madrid, Spain; 4International Exchange Program for Minority Students, Icahn School of Medicine at Mount Sinai, New York, NY 10029, USA; 5Department of Environmental Health, University of Puerto Rico-Medical Sciences Campus, San Juan, PR 00921, USA; 6University Hospital of Guadalajara, 19002 Guadalajara, Spain; 7Department of English Language and Linguistics, Narval School, Cartagena, 30319 Murcia, Spain; 8Department of Environmental Medicine and Public Health, Division of International Health, Icahn School of Medicine at Mount Sinai, New York, NY 10029, USA

**Keywords:** green space, blue space, school, neurodevelopment, neurobehavior, nature, exposure

## Abstract

Today more than half of the world’s population lives in urban areas. Children spend about 40 h a week in the school environment. Knowing the influence of school exposure to green/blue spaces could improve the children’s health, creating healthier environments and preventing exposure to legal/illegal drugs. This systematic review summarized the main results of published studies on active or passive exposure to green or blue spaces in different domains of child neurodevelopment. In August 2022, five databases were searched and twenty-eight eligible studies were included in the analysis. Cognitive and/or academic performance was the most frequently studied (15/28). Most studies evaluate passive exposure to green/blue spaces (19/28) versus active exposure (9/28). Only three studies addressed the relationship between blue space and neurodevelopment. The main results point toward mixed evidence of a protective relationship between green/blue space exposure and neurodevelopment, especially in improving cognitive/academic performance, attention restoration, behavior, and impulsivity. Renaturalizing school spaces and promoting “greener” capacities for school environmental health could improve children’s neurodevelopment. There was great heterogeneity in methodologies and adjustment for confounding factors across studies. Future research should seek a standardized approach to delivering school environmental health interventions beneficial to children’s development.

## 1. Introduction

There is a strong connection between the environment and the state of health [1]. It is estimated that more than 78% of people live in urban nuclei [2]. The impact of urban growth is increasingly felt beyond city limits, and cities have economic, social, and environmental interdependence [3]. Compared to rural areas, urban dwellers are exposed to risks originating from social (e.g., segregation, marginality, and crime) and physical (e.g., urban design, air pollution, and lack of contact with nature) environments that directly impact human health. Making cities and human settlements inclusive, safe, and resilient, and providing universal access to green areas and safe, inclusive, and accessible public spaces, especially for women and children, the elderly, and persons with disabilities, are part of the seventeen Sustainable Development Goals in the 2030 Agenda of the United Nations (UN) [4]. There are many efforts to include green or blue spaces in the architecture of cities; these spaces or their proximity provide ecosystem services, ecological benefits, and recreational, social, and cultural values. Green spaces refer to vegetation (e.g., trees, grass, forests, and parks), while blue spaces are all surface waters visible in the area (e.g., lakes, rivers, and coastal waters). Following the definition of Norwood [5], we can expose ourselves to nature in two main ways, passively or actively. Passive exposure can be understood as that in which the individual is surrounded by nature without direct interaction; it is mainly measured by green exposure methods such as the normalized difference vegetation index (NDVI) or vegetation cover. On the other hand, in active exposure, the participants are immersed in nature and use activities such as outdoor classes, walks, or unstructured games. 

There are various health benefits associated to contact with nature, including better control/prevention of chronic diseases, decreased mortality, improved mental health, greater social cohesion, and reduced health inequalities, among other benefits [6,7]. In childhood, living or playing in natural environments seems to help them to acquire skills, increase their self-esteem and resilience strategies, make them more independent, stimulate more cooperative and creative forms of play, and prevent the use of legal and illegal drugs [8,9]. All these positive childhood experiences in nature influence the community, which promotes pro-environmental behavior from childhood to adulthood [10]. Scientific evidence seems to support the relationship between contact with nature and children’s health, especially at the cognitive, behavioral, or mental level [11]. For this reason, the European Commission recommends that open public green (or blue) spaces be accessible 300 m from the residences [12].

Children are not small adults; they are more vulnerable to environmental hazards than adults [13]. They spend nearly forty hours weekly in schools and colleges [14]. It is in schools where children and adolescents spend more hours after their household. Therefore, schools must be safe places for our children to learn, play, and live free from physical, chemical, biological, and social environmental hazards (see Table 1).

For that reason, creating positive and healthy school environments can have numerous benefits to improve health, well-being, and academic performance and reduce inequalities from the short to the long term [15]. Intervening in schools has a significant scope; the adjusted net school attendance rate is estimated to have reached 87% in 2021 [16]. Schools are complex entities whose operating elements are structural (organizational and physical), social, and cultural factors. These, directly and indirectly, impact students’ health and cognitive development [17]. In a recent guideline, the initiatives for Environmental Health Schools [18] share common objectives, such as promoting knowledge, healthy and sustainable behavior, well-being, resilience, innovation, and developing critical thinking among students and the educative community. During the pandemic, programs that promote outdoor or nature-based education to increase resilience and adaptation have grown to decrease the risk of SARS-CoV-2 transmissions and have reconnected schoolchildren with nature [19]. These initiatives foster new capacities and inter-curricular projects initiatives, where learning takes place in direct contact with nature, such as school gardens, programs such as walking to school or space for gardens, as well as protecting the interior space, improving ventilation, enhancing natural lighting, and increasing visual connection (these activities increase children’s awareness of the concerns and environmental processes). Finally, it is important to accompany these interventions with policies of an ecological approach with practical, healthy, sustainable initiatives maintained over time that covers the interventions carried out at school and/or in the community [20]. The community’s involvement in integrating green spaces in schools is essential, and the school nurse’s role is strategic to unite the triad: environment, education, and health promotion.

**Table 1 ijerph-20-03958-t001:** Basic aspects to be addressed in school environmental health [21].

Groups	Type of Risk
Provision of basic needs	Construction in a safe place (away from roads and avenues, highways, and hazardous industries…)
	Safe building materials
	Adequate temperature
	Water, healthy food
	Light
	Ventilation
	Tobacco-free schools
	Appropriate, non-crowded classrooms
	Safe playgrounds
	Contact with nature
	Sanitary facilities
	Emergency medical assistance
Protection from pollutants and biohazards	Fungi
	Scarce and unsafe water
	Poor food safety
	Vector-borne diseases
	Poisonous and thorny plants
	Poisonous animals
	Rats and dangerous insects
	Other animals (dogs, rodents, insects)
Protection from social pollutants	School and social violence
	Advertising pollution (tobacco, alcohol…)
	Access to junk food
Protection from physical pollutants	Noise
	Extreme heat and cold
	Radiation (radon, ultraviolet, and high-voltage power lines)
Protection from chemical pollutants	Tobacco and alcohol
	Outdoor air pollutants (traffic and transport, industries…)
	Indoor air pollutants (volatile organic compounds, carbon monoxide, heavy metals, laboratory products, spores…)
	Water pollutants
	Pesticides
	Asbestos
	Paints
	Cleaning products
	Hazardous wastes and products
	Diesel particulates in school buses

Collaborative partnerships between healthcare professionals and educational professionals have significant potential to promote ecological health promotion in schools [22]. For these reasons, this review aims to identify the benefits of contact with nature on neurodevelopment in children and adolescents when interacting passively and/or actively in the school environment. 

## 2. Materials and Methods

### 2.1. Search Strategy and Sources of Information

The review was conducted according to the Preferred Reporting Items for Systematic reviews and Meta-Analyses (PRISMA) statement [23]. The search strategy was designed to identify studies relating active and passive exposure to green and blue spaces in school settings with aspects of neurodevelopment, neurobehavior, or both, in children and adolescents. First, a literature search based on the most recent literature in databases such as PubMed, Scopus, Cochrane, GreenFILE, and sciELO was done by using keywords in August 2022. Next, three reviewers searched electronic databases. Studies identified from the various searches were combined, the duplicates were removed, and the articles were reviewed based on their title and abstract content to determine their relevance to the review. New articles were found using the citations of included publications using the “snowball effect” strategy (Figure 1). Different searches were performed in the databases used, connecting with the Boolean operators “AND” and “OR” using the terms Mesh (“Parks, recreational”, “Schools”, “Child development”, “Academic performance”, “Mental health”, “Neurodevelopmental Disorders”, “Neurobehavioral manifestations”) and others (“Green space”, “Blue space”, “Natural outdoor”, “natural spaces” “outdoor space”, “neurodevelopment”, “neurobehavioral”, “neurobehavioral development”) in different combinations (see Appendix A). In addition, a filter was used to obtain results no older than five years (August 2017–August 2022). The search for articles was conducted in English and Spanish, although only articles in English were found.

### 2.2. Inclusion and Exclusion Criteria

Studies were assessed by three review authors and included if (1) there was exposure to green or blue spaces in or around the school; (2) the age of the participants was eighteen years old or less; (3) some type of evaluation measure was used for aspects related to neurodevelopment, neurobehavior or both; (4) the articles were written in English or Spanish. Articles excluded were those that only collected anthropometric data and/or physical measurements, had results not related to the objectives of the review, studies with exposures to green/blue spaces only around the residence or neighborhood, and reviews.

### 2.3. Identification of Studies and Data Collection

The complete manuscripts of all the references that were included as potentially relevant were obtained. The articles that were in doubt of acceptance or rejection were shared among the reviewers for their final inclusion/exclusion decision. Data extracted from each article included authors, year of publication, country; city, study design, sample size, project/study in which it is framed, age at the time of exposure and metric of exposure to green/blue spaces, active/passive exposure to nature, source of exposure data to green/blue spaces, neurodevelopmental/neurobehavioral domain, assessment tool, age at the time of result, method of analysis, adjustment/confounding factors, main results, and limitations/strengths. Active or passive exposure to nature was performed following Noorwood’s definition [5].

### 2.4. Quality Assessment

The quality assessment of the studies included in the review was based on the Quality Assessment Tool with Diverse Studies (QuADS) [24]. QuADS is a tool based on the Quality Assessment Tool Studies with Diverse Designs (QATSDD) of 2012 [25], designed primarily to assess the quality of studies with heterogeneous designs. QuADS is a refined version of QATSDD, and has been shown to have robust psychometric properties and is suitable for both systematic and narrative reviews. The reviewers considered that QuADS demonstrated reliability in content validity and face validity. QuADS evaluate the quality of the studies through 13 criteria related to the content of the publication. Its rating ranges from 0 (lowest quality) to 3 (highest quality). It is not considered a global qualification of the quality of the study since there is no cut-off score that considers the studies to be of high or low quality; the cut-offs would be arbitrary and would not be appropriate. The quality assessment results should be discussed in narrative forms and consider the areas where the information is more or less complete and why. In cases where the ratings differed between the reviewers, each reviewer explained the reasons for their selection. Then, for any remaining discrepancies, the scores of the three raters were averaged.

## 3. Results

### 3.1. Study Selection

One thousand six articles were identified by searching the databases. After eliminating duplicates, nine hundred ninety-three articles remained and were entered into the title and abstract screening phase. At the end of the title and abstract screening phase, thirty-five articles were selected for review. The full texts were reviewed based on the inclusion and exclusion criteria for their final incorporation into the review; eleven articles were excluded, and twenty-four were selected. Figure 1 indicates the reasons for the exclusion of these articles. Four new articles were also incorporated using the snowball effect, after searching the references of the selected previous studies. Finally, a total of twenty-eight articles were included (twenty-four from the database search and four from the reference search) on 494,963 children ranging from 25 [26] to 344,175 [27]. 7983 schools participated, ranging from 2 [28] to 3745 [29]

Of the twenty-eight articles included in this review, 50% (*n* = 14) of the studies were conducted from 2020 onwards. Of the included studies, cross-sectional designs (*n* = 12; 42.86%) were the largest, and the remaining studies were longitudinal (*n* = 8; 28.57%) and experimental/quasi-experimental studies (*n* = 8; 28.57%). Most of the studies with a cross-sectional design were conducted in the United States (six studies), followed by Europe (two studies) and China (two studies). The remaining two were conducted in Brazil and Australia. Of the eight longitudinal studies, four were conducted in the USA, three in Europe, and one in Canada. Eight experimental/quasi-experimental studies were included in this review, most of them conducted in Europe, one in the US, and the last one in Australia.

### 3.2. Quality Analysis of the Included Studies

The total scores for all the studies were high, with an average score of 32.4 and a maximum score of 39 (see Appendix A). The lowest scores were found for item 12, related to the evidence that the interested parties were considered in the design of the research, finding a lack of evidence that suggests that the contributions of the interested parties were considered. The highest scores were given in the items related to the description of the research environment and target population, the adequate design of the study in accordance with the established objectives, and the selected data collection formats and tools. None of the selected studies were considered bad or low quality.

### 3.3. Exposure Assessment

The studies included in this review applied a variety of methods to assess the exposure to green and/or blue space, which can be classified into four (4) main groups: (a) availability of surrounding greenery (speaking in terms of the amount of green space based on different indices, including the NDVI, land use and land cover maps, and vegetation cover maps based on atlases and inventories); (b) accessibility to green and/or blue spaces (access to parks and/or gardens); (c) other indicators related to natural spaces (% of the tree canopy, vegetation inventory, and % of treetops near the road); (d) active interventions (green walls in schools, elements within the classroom, green schoolyard, and outdoor classes in natural settings). In studies evaluating passive exposure, different buffer sizes were used, from 25 m to 2000 m, to measure the amount and availability of green spaces. All studies that used a given buffer size used a circular shape. Most studies did not consider the season (winter, summer, spring, fall) between exposure and outcome assessment. The most used tool to measure passive exposure to green spaces was the NDVI, used in twelve of the nineteen studies that measured passive exposure to green spaces. The rest of the studies that measured passive exposure to green spaces used data from land use and land cover maps based on vegetation atlases or national or regional inventories. For active exposures in nature, outings to nearby natural spaces were used through playful activities or with classes in the open air, playful activities scheduled in the green playground of the center, or activities inside the classroom with natural elements and passive vision of vegetation through the classroom windows. Table 2 shows the type of exposure and the main sources of information.

### 3.4. Outcomes

We identified seven types of outcomes related to different aspects of neurodevelopment and/or neurobehavior, including (ordered by frequency) cognitive and/or academic performance (15), restoration of attention (8), behavior and impulsivity (8), conduct and social interaction (5), neurodevelopmental diseases and disorders (4), working memory (3), and emotional well-being (3). No articles related to motor development (gross/fine) were found. Different methodologies were applied to characterize these results. In general, the most used instruments were reading and mathematics tests scores (*n* = 8), the Wechsler intelligence scale (*n* = 3) for cognitive and/or academic performance, the Bells Test (*n* = 2) for attention restoration, and the Digit Span Memory Test (*n* = 2) for working memory. Below, we report the summary of the results related to each of the neurodevelopmental/neurobehavioral aspects identified in the selected articles. Table 3 presents the general characteristics of the studies included.

#### 3.4.1. Cognitive and/or Academic Performance

Greenness has been positively associated with academic performance in eight studies [27,29,31,32,34,35,39,42]. To evaluate this relationship, most studies focus on punctuation and math and, in some cases, reading level and math [36].

For cognitive performance, one study [44] found a negative and significant association with NDVI in the school surroundings in both the crude and adjusted models. This study used the Wechsler Intelligence Scale for Children—The Third Edition (WISC III), which yields three IQs, a Verbal Scale IQ, a Performance Scale IQ, and a Full-Scale IQ [44]. The different range of tests that constitute the WISC III provides not only a global Intelligence Quotient (IQ) but also a Verbal IQ and a Performance IQ [54]. Regarding greenness, the most used techniques were mapping, satellite-based indices, GIS-based land use variables, and NDVI to evaluate greenness. Nevertheless, it measured different aspects and considered different confusion factors. Five studies [31,34,35,36,42] evaluated the different types of vegetation and their relationship with academic performance. Kweon examines the contributions of different kinds of green cover (tree or grass/shrub cover) on academic achievement in students of 7 to 16 years old in Columbia, US.

Meanwhile, Sivarajah examines the potential effects of tree cover, diversity, and species composition on the academic performance of 8–9 and 11–12-year-old students in Toronto, Canada. Moreover, the study presented a set of plant species that can positively affect children’s academic performance. One study considered how the vegetation and vehicle emissions surrounding primary schools were related to the academic performance of their students in an urban area of Australia [29]. Vegetation within schoolyards and Euclidean buffers (100, 300, and 1000 m) were assessed using the NDVI, and weighted road density was computed for each buffer as a vehicle emissions proxy. Carver found that vehicle emissions were inversely associated with literacy and mathematics scores and mediated some associations of vegetation.

In addition, other authors explored blue exposition [32,44]. The first author studied the exposure to blue spaces and various measures of intelligence quotient (IQ) among children in the Metropolitan Area of Porto (Portugal) but did not find clear associations.

Moreover, the second author used the USGS Hydrographic Cover Dataset and analyzed the relationship between the mean score of the reading test in children (8–9 years old) in Minnesota, US, and found a positive relationship with water coverage, but it was not significant.

#### 3.4.2. Attention Restoration

Studies that attempt to evaluate nature’s effects in the restoration of attention, both selective and sustained, are based on directing attention toward a goal by trying to ignore a series of distractions. Eight studies focused on evaluating this neurobehavioral skill [28,41,45,48,49,51,52,53]. Five studies found that exposure to natural environments, through programmed green play or exercise activities in the yard or nearby green spaces, and even activities with natural elements within the classroom, can positively affect attention control [30,37,48,50,52]. Four of these studies [41,49,51,53] also found positive differences with the control groups, subjected to activities within classrooms or conventionally built spaces, as opposed to activities directly in contact with nature. The activities varied from green exercises during recess in nearby natural spaces to classes in open-air green spaces, and activities with natural elements inside the classroom. The fifth study [45], based on passive exposure to green spaces in adolescents from Flanders (Belgium), found significant associations between the combined vegetation of the residence and the school in a ratio of 2000 m with the reaction time in the Stroop Test and the Continuous Performance Test. The three remaining studies [28,48,52] found no significant differences in exposure to natural spaces and attention control. The first two used active exposure to green spaces through playful activities in samples of primary and secondary school students, respectively. The third study used passive exposure to school green spaces within a large sample but found no relationship between the variables related to attention.

The methodologies used were heterogeneous. Up to six different methodologies were used to measure attention, depending on whether the authors were studying selective attention, sustained attention, or both. The most used method was the Bells Test [41,51] to measure selective and sustained attention, and the Attention Network (ANT) [48,52] to measure selective attention. Both methodologies are based on finding or tracking a target among a series of distractors. Other authors decided to study two types of attention separately, such as Bijnens, who used the Stroop Test (selective attention) based on indicating the color of the written word as soon as possible, or the Continuous Performance Test (sustained and selective attention), which is based on indicating a particular letter within a series of 48 letters as quickly as possible. The d2 Letter Cancellation Test used by Miyrgind followed a methodology similar to the previous test by having to identify the letter “d” with two apostrophes among a series of distractors. Other tests used to measure attention included counting the number of redirections (calls for attention) that had to be given in class, which allowed them to determine the degree of concentration and attention between classes in a natural environment or in conventional classrooms [49,53] as well as the count of children who were “off-task”, that is, children who had been distracted and were not concentrating on the academic task during a determined period.

#### 3.4.3. Working Memory

Three studies [41,51,52] studied the effect of nature in the school environment on working memory performance in children and adolescents. Only one study [52], carried out on children from six European countries who participated in a longitudinal study where multiple variables and environmental exposures were analyzed, did not find significant relationships between school children’s exposure to natural spaces and any of the tests of neurobehavior.

Two of them found a positive effect [41,51] of nature on working memory in primary school children from Lisbon (Portugal) who were introduced to natural elements that were observed from the windows and inside the classroom, and in middle-class children from public schools in Rome (Italy), who conducted outdoor classes in green areas. Both studies used the Digit Span Forward and Digit Span Backward, in which they had to write the digits. In the study performed in Rome, a 30 min active play activity was carried out in a green schoolyard, while in the Lisbon study, there was a window overlooking a 150 × 250 cm artificial green wall and a horticulture intervention where each child had to plant and care for a lettuce plant. These two studies indicate that using control groups, both active and passive nature-related interventions have a positive impact on short-term working memory information retention capacity. 

#### 3.4.4. Emotional Well-Being

Among the three studies assessing students’ emotional well-being, all of them obtained positive associations [33,35,37]. Scott, in his study on children at educational risk, found that increasing tree greenery promotes incremental gains in student well-being, especially at the level of initiative and attachment [33]. The other two articles evaluate interventions. In one article, they were conducted monthly for six months in children with social, emotional, and behavioral difficulties [47], and the other conducted intervention for six weeks (indoor classes and outdoor classes) [53]. The methodologies used were different in the three studies. Scott used data from the Devereux Early Childhood Assessment Preschool Program (DECA-P2) to assess well-being. Chiumento used the Mental Wellbeing Impact Assessment (MWIA) and the Wellbeing Check Cards (WCA) to allow schoolchildren to describe their well-being for themselves. Finally, Largo-Wighs made use of the reports on well-being generated by teachers and students after the intervention.

#### 3.4.5. Behavior and Impulsivity

Eight studies that have tried to understand or relate the behavior of students exposed to natural spaces in the school environment were mainly focused on the study of factors such as aggressiveness, self-control, or impulsivity [33,43,45,47,49,50,51,53]. Four out of eight studies examining neurobehavior and related factors found an improvement in the results. Scott found an improvement in the four spheres of the Devereux Early Childhood Assessment Preschool Program (DECA-P2), which assesses socio-emotional resources, including behavioral aspects, after studying passive exposure to green spaces in students in a North Carolina (US) preschool. Furthermore, Bates found indications that green playgrounds promote environments with less aggressiveness and bullying with more positive social interactions for those students who lived in low-income neighborhoods in Chicago (US). In addition, Ezpeleta found improvements in obsessive-compulsive behavior using the Spence Children’s Anxiety Scale-Parent in those children with larger green spaces around the educational center, especially in girls and those participants with a higher socioeconomic level. Finally, Largo-Wight found that preschool children who conducted their classes in nature had a lower redirection rate than those in indoor classrooms. The remaining studies found no significant associations between passive or active exposure to green environments and behavior. Bijnens found no significant association between total, high, and/or low green space in the school setting, and behavior using the Strengths and Difficulties Questionnaire (SDQ). For their part, Chiumento did not find conclusive results either after analyzing the results of the Wellbeing Check Cards or the Mental Wellbeing Impact Assessment (MWIA) after carrying out an intervention in green spaces specifically designed for each school. Norwood did not find significant changes in student behavior using data from the Composite Index of Classroom Engagement (CICE). However, they did find positive (nonsignificant) changes in student behavior compared to the control group after taking classes in the wild in disadvantaged youth in Queensland (Australia). Finally, Amicone did not find results that suggest that participants who carry out activities in natural spaces increased their impulse control using the Go/No-Go Test. In the control group that carried out the same activity in a built environment, their impulse control did not improve, nor were there differences between the intervention and control groups.

#### 3.4.6. Conduct and Social Interaction

Five studies in our sample evaluated the students’ social interaction and behavior, all having positive associations. Four were related to passive exposure to the school’s greenery, and one was an intervention activity. Surrounding greenery was related to improvements in behavior and self-regulation [33], chronic absenteeism (increasing a one-interquartile range (IQR) of the NDVI decreases absence by 2.6%) [30], and the regulation of obsessive-compulsive disorders [43]. The transformation of greenyards to “ecological greenyards” in neighborhoods, where access to green spaces was limited, also showed high rates of positive or neutral social interactions (and low negative ones), with the impact lasting up to twenty-four months after the intervention [50]. The intervention (based on horticulture) was carried out monthly for six months in a group of thirty-six children who showed improvements in social interactions and the role of the individual in the group [47]. Except for the study on absenteeism, all the others were focused on groups of children with educational, behavioral, emotional, social, economic, or environmental risks.

#### 3.4.7. Neurodevelopmental Diseases and Disorders

Our study included four articles that investigated greenness and neurodevelopment/Neurobehavior, focusing on ADHD, autism, and multiple developmental behavioral syndromes. Two studies focused on autism [26,46]. The first study focuses on whether the experiences and activities in forest schools improve the symptoms of children with autism [26]. This was a quasi-experimental study with elementary school children that found benefits in autistic children through opportunities for play, the exercise of autonomy, and the development of practical, motor, and social skills. The second study assesses whether school districts with more green space have a lower prevalence of childhood autism. A cross-sectional study, in public elementary school districts, found that a 10% increase in the forest, middle tree canopy, and roadside tree canopy would mean a 10%, 11%, and 19% reduction in autism risk, respectively [46]. Additionally, there was one study that focused on ADHD [37]. This study wanted to assess the association between greenery around schools or daycare centers and ADHD symptoms in children. They found that an increase of 0.1 units in the NDVI within 500 m of a school or kindergarten was significantly associated with lower odds of ADHD symptoms. Lastly, one of the studies focused on multiple developmental behavioral syndromes [40]. This study aimed to investigate the associations of exposure to green spaces with multiple behavioral development syndromes in preschool children in China. They performed a cross-sectional study in Wuhan, China, from April 2016 to June 2018. They recruited a sample of 6039 children aged 5–6 years from 17 kindergartens located in five urban districts of the city. They observed a decrease in problem behaviors associated with kindergarten and residence-kindergarten-weighted surrounding greenery in preschool-aged children.

## 4. Discussion

Despite the many hours that children and adolescents spend at school, there are still few studies that evaluate the exposure to the school environment on their health. The impact of nature on the school environment is still understudied compared to residential exposures. The proof is scarce, but this review suggests that there is evidence that contact with nature in the school environment seems to positively influence cognitive and behavioral development in children and adolescents. These results were examined according to the type of intervention (passive/active) and the neurodevelopmental/neurobehavioral domain affected. 

Numerous positive effects were found, especially in the areas of academic/cognitive performance. The simple fact was that conducting classes outdoors in natural environments could influence and improve the students’ cognitive level in the short term, especially in adolescents subjected to a strong dose of stress due to academic performance. Only two studies found negative effects between the level of vegetation and student achievement [36,44]. Four found no relationship between exposure to nature and the neurodevelopmental/neurobehavioral domains on which they investigated. These results suggest that natural exposures could improve key neurodevelopmental processes related mainly to cognitive performance, attention restoration, behavior, and impulsivity. It was not dependent on what the intervention was whether it was passive or active. Given that all included studies had a relatively high-quality score on the QuADS (*n* = 28, ≥25 points), the conclusions remain robust and have practical implications for school-age children and adolescents. 

This review explored the intrinsic factors of green and/or blue spaces that may produce more favorable cognitive outcomes in school-age children and adolescents. Among the environmental variables related to green spaces, the canopy or tree cover explained better results in neurodevelopment [31,32,33,34,35,42,43,45,46], such as cognitive performance, than other green covers, such as low-growing or herbaceous vegetation. These findings indicate that the initiatives aimed at improving the physical space of the school environment with vegetation should be focused on plantations with tall vegetation, such as trees, which could evolve into greater cost-effectiveness for health than other types of low vegetation, such as grass. Other authors studied confounding factors or environmental mediators, such as air pollution [29,30,45], finding that although air pollution occurred as a negative factor in the results of the neurodevelopmental evaluation methodologies, the nearby vegetation acted as a mediator of these negative effects, especially with the closest vegetation (<100 m) to the educational centers. However, many studies have yet to evaluate some of these factors better to understand the results of the relationships between their variables. It is considered necessary to include more confounding or mediating factors (toxic habits, physical activity outside school hours, place of residence, etc.) regarding the environmental and social variables that favor the understanding of the benefits of nature in combination with factors that promote positive implications for practice and desirable outcomes in cognitive functioning on children and youth well-being. 

This review allowed us to differentiate between nature’s active or passive role in these exposures. Most articles (*n* = 19) studied the passive effect of natural spaces on the health of schoolchildren through vegetation indices and percentages of vegetation cover. This type of study allows carrying out other studies over time to include larger population samples due to the accessibility and use of the data. The accessibility and handling of geographic information systems (GIS) computer tools provide a reliable and verifiable source of information. However, this methodology makes it difficult to work at an individual level or on a smaller scale. Some factors that would help better understand the associations, such as the vegetation types, can be blurred among large results. However, some authors, such as Sivarajah and Ezpeleta, have tried to fill this need for more information with data from inventories or regional vegetation atlases. The active role of nature was studied in nine out of the twenty-eight articles (32%) included in this review. Most were based on activities outside the classroom using natural spaces inside or outside the educational center as a physical location. These studies allow us to know results at an individual level and develop strategies to promote contact with nature that can extrapolate to other school spaces. However, it has some limitations, such as small sample sizes. As Bates and Amicone have shown, short-term school environmental health interventions can play an essential role in restoring attention, working memory, and behavior, while in long-term studies, the effects may be more visible on cognitive and/or academic performance, as has been demonstrated in longitudinal studies carried out by Leung et al. and Almeida et al. Children face increasing cognitive demands, and exposure to nature effectively achieves better results, varying for each age group in different ways.

Another differentiating aspect studied in this review was the evaluation of exposure to blue spaces. Most studies evaluating the benefits of contact with nature on children’s health are directed at green spaces, causing a knowledge gap on the influence of blue health. Only three studies included exposure to blue spaces as a possible factor that could affect health, and only Hodson et al. found a positive relationship (not significant) with the results of academic performance for the mean scores on the reading test. These differences could be due to climatic differences and accessibility to blue spaces derived from the geographical location of the educational centers in the territories studied. In the study by Julvez et al., up to six different European countries were included, where the accessibility and direct interaction with blue spaces can differ significantly, as it could be between Mediterranean countries of Spain or Greece, compared to Nordic and Baltic countries, such as Norway and Lithuania. Exposures to blue environments remain to be explored, as well as the underlying mechanisms that influence these relationships. A recent systematic review [55] on the potential health effects of blue spaces found six studies that evaluated these interactions on children and youth populations, finding improvements in mental and physical health in most of them. For this reason, it is considered necessary that the focus on blue health be included through nature-based activities in school environments through aquatic activities or passive measurement, with the school spaces being coastal or close to inland water bodies (rivers and/or lakes) being the most benefited. 

The pathophysiological mechanisms that could explain the health benefits attributed to green and blue spaces are poorly understood. It seems that multiple and unspecific pathways achieve the effect [56]. Despite this, steps are beginning to be taken in their study, and despite the lack of knowledge, some evidence is beginning to appear. Metainflammatory mechanisms have been proposed through neuroendocrinological effects of stress and attention, variations in the immune system response through (a) stimulation of a microbiome specific to natural spaces or their interaction with volatile organic compounds emitted by the surrounding vegetation, or in the case of blue spaces, due to the aerosols generated by the tide and the sea breeze [57,58]. Some studies following this trend have found changes in salivary cortisol and an increase in cellular and humoral immunity related to the natural killer cells (NK) [59,60]; (b) intrinsic qualities of green and blue spaces that improve health or well-being (restoration theory) and that have a direct or indirect effect, either through the simple visualization or observation of green or blue spaces; (c) the cushioning and healthy effect associated with green spaces (it cushions the impact of temperature, air pollutants, and noise; (d) the opportunity to engage in physical activity; (e) improve social interactions [61]. In addition, related to neurodevelopment, a controlled experiment found associations between experiences in nature and reductions in rumination and decreased neural activity in the subgenual prefrontal cortex, which was related to risks of mental illness, which directly affects infant neurodevelopment [62].

Despite the heterogeneity of the results, we should value the positive findings in the field of green or nature-based interventions related to school environmental health. This work could provide diverse social and political implications for different social areas. At the policy and organizational levels, it can provide evidence to support “green” policies and guidelines related to the design and emplacement of schools in different communities, as well as practical guidelines for renaturalizing already built school environments. In addition, it could promote more sustainable management, which incorporates environmental criteria into school management contracts. At the health level, the evidence of the physical and neurodevelopmental benefits previously described should be meant as a promotion of the school nursing’s figure. They will promote the development of activities in nature in collaboration with teachers, as well as with the school’s parents’ associations. In addition, the protective role of nature has special benefits when it comes to preventing pathologies associated with environmental pollution and the consumption of legal and illegal drugs, two of the most influential factors in children’s health today [63,64]. In relation to drugs, there is a growing body of research suggesting that residential greenness exposure may have protective effects related to substance use [65], which is why we consider that it could be extrapolated to school greenness exposure. The health prescription of nature by health professionals will be indispensable in the future, creating a new way to support the health of people and nature and promoting the concept of “One Health”. Research in this field is still scarce and uncertain; there are numerous avenues of research to deepen the benefits of nature in human health where many variables converge, leaving information gaps. It is essential to continue providing scientific evidence on how the influence of exposure to nature is related to our health, socially, physically, and mentally.

Therefore, exposure to natural spaces correlates with better cognitive and behavioral development of children and adolescents in different ways depending on their level of development. Although the review indicates that exposure to nature benefits throughout childhood, more attention should be paid to early childhood stages, from pregnancy to 5 years old, when 90–95% of human brain weight is built [66]. This time is a period of special vulnerability and opportunity to achieve optimal child development that will accompany them throughout their lives. Along with the physical and mental health benefits, it will allow the development of a pro-environmental and community character that will trigger healthy adults, awareness, and commitment to the environment and their health. Furthermore, identifying specific characteristics of green and blue spaces adapted to each population and climatic context will mean optimizing the efforts of the actions. More concrete descriptions of exposure to nature are required to understand the mechanisms underlying these interactions. In addition, measures related to neurodevelopment and/or neurobehavior must be sensitive and reliable enough to detect significant differences in the age range of children and adolescents, so that the development of nature exposure strategies can be adapted to each age group. 

### Strength and Limitations

The strengths of our review were the large number of databases used for the search (5) and the search for articles in two languages (English and Spanish). Additionally, it provides a global vision of the results and methodologies used that open new research fronts in this field and provide tools for administration managers. In addition, it organizes the exposures into passive or active and presents the outcomes organized by neurodevelopmental domains. The strengths of the studies are related to the inclusion, in a significant part of the studies, of environmental variables, such as vegetation type, pollution, or accessibility, to better explain and understand the results. The large sample sizes were used primarily to draw more robust conclusions. Both qualitative and quantitative measurement tools were used. They allow the opening of future lines of research on the links between natural spaces and children’s health and provide valuable tools for administrations and community managers.

The limitations of our review are due to the high heterogeneity of the methodologies used, which makes it difficult to compare them and extract extrapolated data. There is difficulty and heterogeneity in measuring outcomes related to nervous system functions. The diversity of types of exposure to green and/or blue spaces also presents a limitation, as well as the lack of a clear definition of “contact with nature” in terms of variety, frequency, and time. Additionally, the absence of MeSH terms that homogeneously collected important search terms, such as blue spaces, is a limitation. The main limitation of the reviewed studies that comprise the review is that existing research requires more confounding factors, which limits the quality of the available evidence. In addition, although some of the articles incorporate different environmental or social factors, many others need to be contemplated, and the results may be biased. Access to green and blue spaces should be evaluated to better understand the results. The period in which environmental measurements are made, especially in cross-sectional studies, is vital to understand the relationships between greenness and the results of the methodologies used; we may be able to find different results depending on the season of the year in which it is performed. Other limitations detected are related to the recruitment of the samples, finding biases at the socioeconomic and health level when selecting population samples with particular social and health characteristics. However, the novel and growing nature of this type of study means that the tools used are still so diverse and different, which makes it possible to obtain a wide range of results and conclusions but hinders their reproducibility and comparability. 

## 5. Conclusions

Although there is still limited evidence, the scientific literature seems to find beneficial effects of exposure to natural environments at school, both active and passive, on neurodevelopment (e.g., academic performance, attention restoration, behavior, social interaction, and well-being) of children/adolescents. The development of adult occupational health during the 20th century brought about important changes in the workplace. Children spend about 40 h a week in the school environment. School environmental health is “theoretically children occupational health” and one of the most important health challenges for the 21st century. A redistribution of resources to increase both the development and creation of new green or blue areas or to renaturalize school spaces and promoting of “greener” activities in the school environment and curriculum will be paramount. In addition, it will be necessary to provide training in environmental health to incorporate new jobs such as environmental nurses and scholar environmental specialists.

This systematic review could have various future implications at a political, environmental, health, and scientific level. It is recommended to continue researching the benefits of contact with nature at school and searching for international standardization of health interventions, as it can be an effective tool to promote the health not only of children and adolescents but also of the community.

## Figures and Tables

**Figure 1 ijerph-20-03958-f001:**
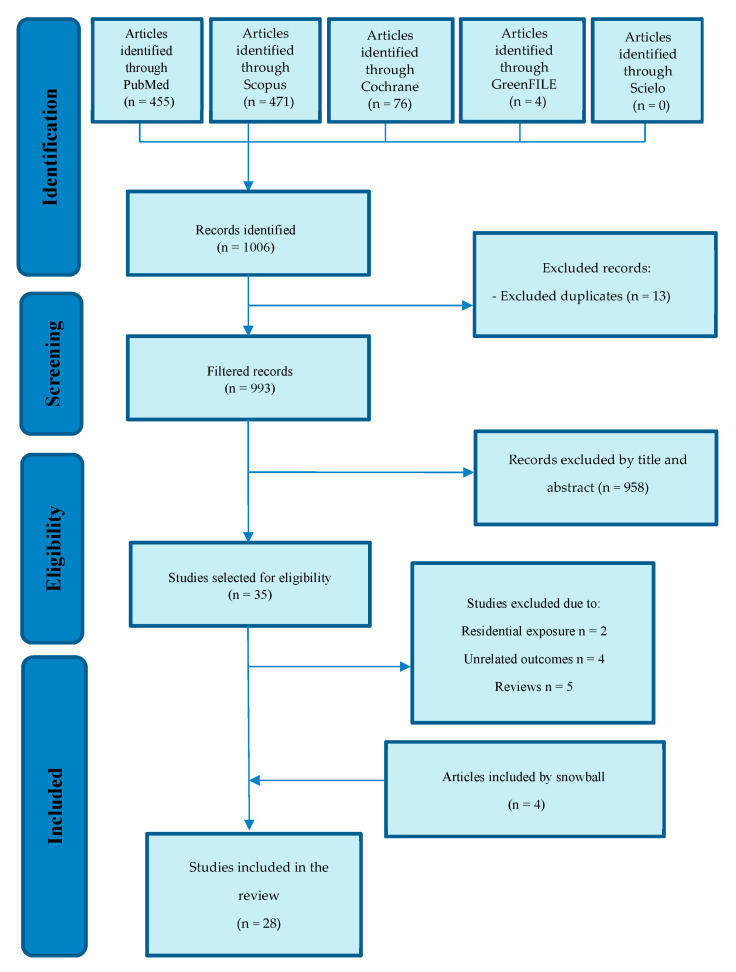
PRISMA-P flowchart of the stepwise assessment of articles obtained from the search strategy.

**Table 2 ijerph-20-03958-t002:** Detailed exposure assessment methods and exposure data sources in the included studies.

Citation	Type of Exposure	Source of Exposure Data
Macnaughton et al. (2017) [30]	School (buffer of green areas at 250 m and 1000 m from each school)	NDVI from MODIS images (NASA)
Kweon et al. (2017) [31]	School (% of trees, grass/shrubs, water, buildings, soil bare and paved surfaces in each school plot)	Land use and cover map (LULC) from high-resolution aerial multispectral LiDAR images
Hodson et al. (2017) [32]	School (% of green or blue coverage in the school’s attendance area)	National land and hydrographic vegetation cover map USGS CoverDataSet
Scott et al. (2018) [33]	School and residential (% tree canopy and impermeable surface in the areas defined around the school (NPA), access to parks in the area)	High-resolution aerial photography (GIS and QOL Study)
Sivarajah et al. (2018) [34]	School (m^2^ of plant surface within the center educational; total land area (m^2^), grass/shrub area (m^2^), tree canopy cover (m^2^), % tree coverage vs. ground area)	Tree canopy assessment data (UTC) and TDSB’s NeighborWoods tree inventory
Kuo et al. (2018) [35]	School (green areas inside the school and in a 25 m buffer zone around the center and attendance area of a neighborhood school)	LiDAR images and program data from National Images Agricultural (NAIP) and tree canopy assessment data Chicago urban (C-UTC)
Browning et al. (2018) [36]	School (green buffer area at 250, 500, 1000, and 2000 m from each school)	NDVI from MODIS remote sensing images from NASA
Yang et al. (2019) [37]	School (green buffer area at 100 m, 500 m, and 1000 m around each school)	NDVI and soil-adjusted vegetation index to the ground (SAVI) from Landsat 5 Thematic Mapper satellite images obtained with a 30 × 30 m resolution
Markevych et al. (2018) [38]	School and residential (green buffer area of 500 m and 1000 m around the house/school)	NDVI from MODIS images (NASA) satellites. Vegetation level and tree cover density (EEA), % of land agriculture, forests, and green spaces around the residence/school (data from the Bavarian Office of Studies)
Leung et al. (2019) [39]	School (circular buffers from 250 to 2000 m around school)	NDVI from MODIS images (NASA) satellites
Liao et al. (2020) [40]	School and residential (circular buffers of 100 m that surround the central point of residences and kindergartens)	NDVI from MODIS images (NASA) satellites
Bernardo et al. (2021) [41]	School (school intervention)	(a) Artificial green wall applied to the classroom window; (b) activity with lettuce pots in the classroom
Kuo et al. (2021) [42]	School (tree canopy cover and canopy cover, total green with circular buffers of 250 m and 1000 m around school)	NDVI from images of the Landsat data of the Service States Geologic United. Tree canopy cover is calculated using the product Service Tree Canopy mapping States Forestry United Database national coverage (NLCD) of 2011, based on multispectral images from Landsat satellites and other terrestrial information and auxiliary available
Ezpeleta et al. (2022) [43]	School and residential (buffers of green areas of 100, 300, and 500 m, coverage trees, and access to green spaces)	NDVI from Landsat images for the Service 2015 US Geological. The vegetation continuum field (VCF) was obtained from Landsat images of the land cover land use change (NASA). Access to green spaces was obtained from the Urban Atlas of the Copernicus Land Monitoring Service for 2012 (European Union, 2020) and the distance Euclidean to each park from the school
Almeida et al. (2022) [44]	School and residential (green buffer area to different sizes around the school, distance, existence, and number of green spaces downtown urban educational and distance to blue spaces)	NDVI from images satellite. The cartography was obtained from digital maps provided for the different area councils of Porto Metropolitan
Requia and Adams (2022) [27]	School. (buffers green buffer area (NDVI) and distance to green spaces at 500 m, 750 m, and 1000 m and number of green spaces)	NDVI from spectroradiometer resolution imaging moderate (MODIS) (NASA). Land use data and the number of green spaces. It was provided by the Institute Brazilian Geography and Statistics Institute Brazilian Geography and Statistics (IBGE)
Bijnens et al. (2022) [45]	School and residential (green buffer area at 50, 100, 300, 500, 1000, and 2000 m, proximity to green spaces, air pollution, and noise)	Data on green coverage space were obtained from the Green Map of Flanders 2012 of the Flemish Geographical Information Agency (AGIV). The data on the proximity of spaces were obtained from a Flemish Government—Department Environment map. The contamination data were obtained from the Flanders Environment Agency (VMM) and pollution data from monitoring stations fixed
Wu et al. (2017) [46]	School (green coverage inside each school, school district, and tree cover around the center and highways)	Data on terrestrial coverage (National Land Cover Dataset (NLCD)), the average height of the average crown of the trees per pixel (30 × 30 m) (cartographic canopy NLCD), % of tree canopy near the road (50 m around roads) (NavTEQ data)
Chiumento et al. (2018) [47]	School (school intervention)	Exposure to green space designed inside the school for the project with sessions monthly based on the intervention “Haven Green Space”
Anabitarte et al. (2021) [48]	School (school intervention)	Exposition to green or gray spaces close to school through playful activities
Norwood et al. (2021) [49]	School (school intervention)	Exhibition in a standard indoor class or in a “green” outdoor classroom
Bates et al. (2018) [50]	School (school intervention)	Exposure to a “green” schoolyard created inside the educational center
Mygind et al. (2018) [28]	School (school intervention)	Active exposure to natural spaces close to the school environment and within the conventional educational classrooms
Friedman et al. (2022) [26]	School (school intervention)	Exposure to nature (trees and pond) in a forestal school with outdoor activities
Carver et al. (2022) [29]	School (NDVI buffer at 100, 300, and 1000 m from the school center and air pollution)	NDVI from NASA LANDSAT 8 Satellite images and raster images. Environmental pollution from proxy using weighted road density (WRD)
Amicone et al. (2018) [51]	School (school intervention)	Exposure to the effect of recreation in a nearby natural space (vs. built space) during school hours both in the morning and afternoon
Julvez et al. (2021) [52]	School and residential (green/blue buffer area at 100, 300, and 500 m from each school and residence)	NDVI from satellite images from Urban Atlas (2006) and EUNIS (2009). Green/blue from Urbanatlas (2006) and EUNIS (2009)
Largo-Wight et al. (2018) [53]	School (school intervention)	Exposure to a standard indoor classroom versus an outdoor classroom in nature

**Table 3 ijerph-20-03958-t003:** Main characteristics of the included studies in the review.

Citation and Geographic Location	Design, Sample Size, and Population	Exposure	Neurodevelopmental Domain (s)	Assessment Tools	Relationship between Variables	Main Findings
Macnaughton et al. (2017) Massachusetts, United States [30]	Cross-sectional study N = 1772 public schools Children and adolescents (6–17 years)	Passive exposure (NDVI; atmospheric pollution)	Conduct and social interaction	Truancy data	(+) Positive and significant	A one-interquartile range increase in NDVI was associated with a 2.6% lower rate of chronic absenteeism (*p* < 0.0001). 23.3% of absenteeism variability is explained by NDVI, PM2.5, race and household income (*p* < 0.05).
Kweon et al. (2017) Columbia, United States [31]	Cross-sectional study N = 219 public schools Students from 2nd to 10th grade (7–16 years old)	Passive exposure (coverage percentage)	Cognitive and/or academic performance	Achievement in reading and mathematics	(+) Positive and significant	Tree cover has a significantly positive association with student performance in mathematics (β = 0.23; *p* < 0.05) and reading (β = 0.22; *p* < 0.05) after adjusting for confounding factors.
Hodson et al. (2017) Minnesota, United States [32]	Cross-sectional study N = 222 urban schools 3rd-grade students (8–9 years old)	Passive exposure (coverage percentage; blue space)	Cognitive and/or academic performance	Achievement in reading and mathematics	(+) Positive and significant	Tree canopy is positively associated with reading level performance (β = 0.1211; *p* < 0.05). The relationship between the mean score of the reading test and the % water coverage was positive (β = 0.0609) but not significant (*p* < 0.10).
Scott et al. (2018) North Carolina, United States [33]	Longitudinal study (6 months) N = 1551 students Preschool students (4–5 years)	Passive exposure (coverage percentage; access to green areas)	Emotional well-being Behavior and impulsivity Conduct and social interaction	DECA-P2	(+) Positive and significant	Students improved approximately 3 more points on the initiative for every 10% reduction in impervious surfaces in the school (γ = −0.30; *p* < 0.05). Children improved almost 1 point in initiative with each 10% increase in access to parks (γ = 0.07; *p* < 0.05). Student self-regulation was estimated to improve by about 1.4 points more with each 10% increase in school tree cover (γ = 0.14; *p* < 0.05). Students improved by 1.6 points in behavioral concerns for every 10% increase in tree cover at the school (γ = −0.16; *p* < 0.05). Students from low-canopy homes who attended high-canopy schools improved more than students exposed to low-canopy at both home and school (by approximately 1.5 points).
Sivarajah et al. (2018) Toronto, Canada [34]	Longitudinal study (4 years) N = 387 primary schools 3rd-grade (8–9 years) and 6th-grade (11–12 years) students	Passive exposure (coverage percentage)	Cognitive and/or academic performance	LOI; achievement in reading and mathematics	(+) Positive and significant	Tree cover was positively related to writing (b = 16.25; *p* = 0.03). Tree cover correlated with all highly challenged 6th grade scores on the LOI (R = 0.229; *p* < 0.05). No differences were found between the % of soft surfaces (shrubs and herbaceous) versus paved. Tree species diversity and relative abundance of conifers had no detectable effect on academic performance.
Kuo et al. (2018) Chicago, United States [35]	Cross-sectional study N = 318 public schools in low-income neighborhoods 3rd-grade students (8–9 years old)	Passive exposure (coverage percentage)	Cognitive and/or academic performance	Achievement in reading and mathematics	(+) Positive and significant	Tree cover in schools was significantly related to reading (R = 0.37; *p* < 0.001) and math (R = 0.35; *p* < 0.001) results. School “greenness” was a better predictor of achievement in math vs. neighborhood greenery (R = 0.37 vs. R = 0.35; *p* < 0.001). Grass/shrub cover was not related to reading or math performance (*p* > 0.05). School tree cover in extremely deprived schools was about half that in less deprived schools (54%).
Browning et al. (2018) Chicago, United States [36]	Longitudinal study (2006–2012) N = 404 public primary schools 3rd-grade students (8–9 years old)	Passive exposure (NDVI)	Cognitive and/or academic performance	Achievement in reading and mathematics	(−) Negative and significant	The fitted model showed significant negative (range −0.051 to −0.027) relationships between greenness and test scores (*p* < 0.01). There is no convincing evidence of a positive relationship between green space and academic performance. The relationship between green space and academic performance may be non-existent or slightly negative in low green space density and high disadvantage settings.
Yang et al. (2019) Liaoning, China [37]	Cross-sectional study (2012–2013) N = 59,754 children from 94 educational centers and nurseries Students aged 2–17	Passive exposure (NDVI)	Neurodevelopmental diseases and disorders	DSM-IV; C-ASQ	(+) Positive and significant	Higher levels of greenery within the first 100m in NDVI and SAVI were associated with lower odds of ADHD symptoms (OR = 0.92 (0.89–0.97; *p* < 0.001) (OR = 0.90 (0.83–0.95); *p* < 0.001) adjusted for age, sex, parental educational level, household income, household district type, and dog ownership. A 0.1 unit increase in the NDVI or soil-adjusted vegetation index within 500 m of a school or kindergarten was significantly associated with lower OR of ADHD symptoms (OR = 0.87 [95% CI, 0.83–0.91] and (OR = 0.80 [95%CI, 0.74–0.86], respectively; *p* < 0.001 for both).
Markevych et al. (2018) Munich and Wesel, Germany [38]	Longitudinal study (2005–2009 and 2011–2014) N = 1351 children from Munich and 1078 from Wesel Students aged 10 and 15 from primary and secondary schools	Passive exposure (NDVI; coverage percentage)	Cognitive and/or academic performance	Achievement in reading and mathematics	(=) No associations	No associations were observed between any of the green space variables and Wesel scores in the children. Several statistically significant associations were observed with German reading and mathematics scores in the Munich children; however, the associations were inconsistent across the sensitivity analysis.
Leung et al. (2019) Massachusetts, USA [39]	Longitudinal study (2006–2014) N = 27,493 students Public school students from 3rd to 10th grade	Passive exposure (NDVI)	Cognitive and/or academic performance	MCAS	(+) Positive and significant	They found a significant positive association (*p* < 0.05) between the greenness of the school environment and academic performance based on % of green areas, after adjusting for possible confounding factors. Greater exposure to green space use areas was also significantly associated with higher academic performance. The positive relationship between the greenness of the school environment and academic performance was constant in the different subpopulations.
Liao, Jiaqiang, et al. (2020) Wuhan, China [40]	Cross-sectional Study (2016–2018). N = 6039 children from 5 to 6 years of age from 17 kindergartens located in five urban districts of the city	Passive exposure (NDVI)	Neurodevelopmental diseases and disorders	CBCL	(+) Positive	A one-interquartile range increase in kindergarten and residence–kindergarten weighted NDVI was associated with decreased T scores for total behavior by 0.61 [95% CI, 1.09–0.13] and 0.49 [95% CI, 0.85–0.12]. Stratified analyses indicated that associations of exposure to green spaces with problem behaviors were stronger in boys than girls.
Bernardo, Fatima, et al. (2021) Lisbon, Portugal [41]	Experimental study N= 95 students Students from 1st to 3rd grade of primary schools	Active exposure (green wall and activities in the classroom)	Cognitive and/or academic performance Restoration of attention Working memory	DGS; WISC; Bells Test	(+) Positive and significant	Results showed a significant increase in sustained and selective attention (T3: (33.0 vs. 29.3); *p* < 0.001) and working memory (6.64 vs. 5.62); *p* < 0.005) between the experimental and control groups, especially at the third moment.
Kuo, Ming, et al. (2021) Washington State, USA [42]	Cross-sectional study. N = 49,255 students 6th-grade primary school students	Passive exposure (NDVI; coverage percentage)	Cognitive and/or academic performance	Achievement in reading and mathematics	(+) Positive and significant	Six of eight spatial error models showed positive and statistically significant relationships between school greenness and sixth-grade student performance. Greenness (250 m and 1000 m)–achievement correlations are consistently larger for tree canopy than for total greenness (0.400 vs. 0.330; 0.372 vs. 0.322; 0.376 vs. 0.330; 0.337 vs. 0.293; *p* < 0.05).
Ezpeleta et al. (2022) Barcelona, Spain [43]	Cross-sectional study N = 378 children Boys and girls of 9 and 10 years of age	Passive exposure (NDVI; access to green areas)	Behavior and impulsivity Conduct and social interaction	SCAS-*p*	(+) Positive and significant	Linear and mixed effects models showed that green spaces at school, but not at home, were significantly associated with reduced obsessive-compulsive behavior in buffer zones, with benefits for girls as well as boys with graduated parents. Higher green spaces around the school might be associated with less obsessive-compulsive behavior in elementary school children, especially girls and those of higher socioeconomic status.
Almeida et al. (2022) Porto, Portugal [44]	Longitudinal study (2005–2015) N = 3827 children Birth cohort of the XXI generation (G21) boys and girls up to 10 years of age from birth	Passive exposure (NDVI; access to green and blue areas)	Cognitive and/or academic performance	WISC	(−) Negative and significant	The NDVI in the surroundings of the school had a negative and statistically significant association. In the adjusted model, the association remained for a distance of 50 m and performance IQ −12.70 [95%CI, −25.03 (−0.48)] and for a distance of 100 m and verbal (−16.52 [95%, −30.33 (−2.60)]), performance −12.99, [95%, −25.72 (−0.40)] and global IQ −16.59 [95%, −30.33 (−2.84)]. No clear associations were observed regarding accessibility to blue spaces. Physical activity seemed to have a minor mediating role.
Requia and Adams (2022) Federal District, Brazil [27]	Cross-sectional study. N = 344,175 students (256 schools) Students in middle school, high school, and adult learning programs	Passive exposure (NDVI; access to green areas)	Cognitive and/or academic performance	Academic qualifications	(+) Positive and significant	They estimated that NDVI is positively associated with academic achievement at the school level, with an estimated coefficient of 0.91 [95%CI, 0.83–0.99] for NDVI values at the centroid of a school. Distance to green areas was negatively associated with academic performance (−2.09 × 10^−5^ [95%CI, 3.91 × 10^−5^ (−2.84 × 10^−6^)]. The number of green areas was estimated with mixed results (direction of the association), depending on the size of the buffer.
Bijnens et al. (2022) Flanders, Belgium [45]	Cross-sectional study N = 596 adolescents Students between 13 and 17 years of age	Passive exposure (coverage percentage; access to green areas; atmospheric pollution)	Restoration of attention Behavior and impulsivity	STROOP; CPT; SDQ	(+) Positive and significant	An IQR (13%) increment in total green space within 2000 m of the residence and school combined, is associated with a 32.7 ms [95%CI, −58.9 (−6.5)]; *p* = 0.02) and a 7.28 ms (95%CI, −11.7 (−2.8)]; *p* = 0.001) shorter mean reaction time between the presentation of a stimulus and the response based on the Stroop Test and the Continuous Performance Test. Green space higher than 3 m is associated with a faster reaction time of the Continuous Performance Test −6.50 ms; ([95%CI, −10.9 (−2.2)]; *p* = 0.004), while low green is not.
Wu et al. (2017) California, United States [46]	Cross-sectional study N = 543 public elementary school districts Students from 5 to 12 years of age in elementary schools and special education	Passive exposure (coverage percentage)	Neurodevelopmental diseases and disorders	Autism prevalence from CASEMIS	(+) Positive and significant	Autism rate was inversely associated with forest (RR = 0.96 [95%CI, 0.93–0.99]) and average tree canopy (RR = 0.96 [95%CI, 0.92–0.99]). Urban area and road density were positively associated with the rate of autism (RR = 1.09 [95%CI, 1.06–1.10]) (RR = 1.21 [95%CI, 1.14 −1.27]). A 10% increase in the forest, middle tree canopy, and tree canopy near the road would mean a reduction in risk for autism of 10, 11, and 19% respectively (*p* > 0.001).
Chiumento et al. (2018) North West England [47]	Quasi-experimental study N = 36 children with behavioral, social, and emotional difficulties from 3 different schools Students aged 9–15 years	Active exposure (green schoolyard)	Emotional well-being Behavior and impulsivity Conduct and social interaction	MWIA; WCA	(+) Positive	There were no statistically significant results in the analysis of the results of the well-being checking cards. However, some variables such as “the feeling of friendship” improved remarkably in the 3 schools. MWIA factors related to mental health and well-being such as “emotional well-being” and “self-help” were positively impacted by the interventions.
Anabitarte et al. (2021) Basque Country, Spain [48]	Experimental study N = 167 children from 4 primary schools 7-year-old students from primary schools	Active exposure (activities in nearby natural spaces)	Restoration of attention	ANT	(=) No associations	No attention restoration effects were found after performing the interventions, neither in the green nor in the gray space. No differences were found between the groups that carried out the activity in the green space near the school and those that carried it out in the gray space near their school.
Norwood et al. (2021) Queensland, Australia [49]	Experimental study N = 73 students from 3 different classes of the same school 13–14-year-old students in secondary education	Active exposure (outdoor classes in nature)	Cognitive and/or academic performance Behavior and impulsivity Restoration of attention	CICE; redirects; academic qualifications	(+) Positive	No significant changes were found in the behavior or learning of the students (*p* > 0.05). Positive changes (not significant) were found in the behavior of the students in carrying out the task compared to the control group. Outdoor classrooms required fewer reroutes than indoor classrooms (on average, indoor classrooms that would produce 60 redirects in the same outdoor classroom would produce 45). Outdoor classrooms may promote greater engagement and better on-task behavior than their indoor counterparts, but this does not always turn into better grades.
Bates et al. (2018) Chicago, United States [50]	Longitudinal study (2016–2017) N = 3 low-income public primary schools Students from kindergarten to 8th grade	Active exposure (green schoolyard)	Behavior and impulsivity Conduct and social interaction	Reports from teachers and tutors (behavioral and physical activity); CARS	(+) Positive	The students from green schoolyards maintained high rates of positive (between 27.10 and 35.20% depending on the season) or neutral social interactions and very low rates of negative interactions (between 2.50 and 2.80% depending on the season). Green schoolyard favors more safety at recess (t = 1.21–1.24; *p* < 0.005) and a decrease in bullying (t = 0.53; *p* < 0.005) measured by teachers.
Mygind et al. (2018) Copenhagen, Denmark [28]	Quasi-experimental study N = 47 schoolchildren from 2 primary schools Students from 10 to 12 years of age	Active exposure (outdoor classes in nature)	Cognitive and/or academic performance Restoration of attention	HR; d2 Test	(=) No associations	There was no evidence of superior cognitive performance in natural settings compared to classrooms.
Friedman et al. (2022) East of England [26]	Quasi-experimental study N = 25 primary school children participating in forest schools (FS) Students with autism from 8 to 12 years of age	Active exposure (outdoor classes in nature)	Neurodevelopmental diseases and disorders	Reports from children and parents (behavior and well-being)	(+) Positive	FS benefits autistic children through play opportunities, the exercise of their autonomy, and the development of practical, motor, and social skills. FS sessions provide autonomy to children with autism.
Carver et al. (2022) Australia [29]	Cross-sectional study N = 3745 primary school students of 8 or 10 years of age in primary schools	Passive exposure (NDVI; atmospheric pollution)	Cognitive and/or academic performance	Achievement in reading and mathematics	(+) Positive and significant	The highest positive associations of NDVI with academic performance scores occurred in reading (b = 60.45; *p* < 0.001) for grade 3 at 100 m distance; and in reading (b = 32.33; *p* < 0.001) for the 5th grade at a distance of 300 m. Vegetation around primary schools is associated with higher performance in reading and mathematics, partially mediating vehicle emissions, with the highest scores in grade 3 for mathematics at 100 m distance (b = 37.01; *p* < 0.001). Vehicle emissions as a mediator, in particular, of the associations of nearby vegetation (<300 m from schools) with reading (grade 5), mathematics (grade 3), and language conventions scores (5th grade), representing up to 37% of these associations.
Amicone et al. (2018) Rome, Italy [51]	Quasi-experimental study N = 118 4th- and 5th-year primary school students from urban areas	Active exposure (recreation in nearby natural spaces)	Restoration of attention Working memory Behavior and impulsivity	Bells Test; DGS; WISC; Test Go/No- Go	(+) Positive and significant	Greater increase in sustained and selective attention in the natural environment (T1: (M = −0.08; SD = 1.21)) (T2: (M = 0.37; SD = 1.10)) compared to the built environment (T1: (M = 0.102; SD = 0.78)) (T2: (M = −0.40; SD = 0.72)). The perceived restoration also showed significantly higher positive results in the natural environment (M = 5.33; SD = 2.63) compared to the built environment (M = 2.85; SD = 1.71). Participants in the natural setting reported a significant improvement in digit-rank test scores at T1 (M = 15.22; SD = 0.34) to T2 (M = 16.38: SD = 0.38). Standardized playtime team play and individual free play recess in a natural (vs. built) environment favor restoring student attention both during morning and afternoon school hours.
Julvez et al. (2021) 6 European countries (United Kingdom, France, Spain, Lithuania, Norway and Greece) [52]	Longitudinal study N = 1298 mother–child pairs from different European cohorts Children aged between 6 and 11 years old	Passive exposure (NDVI; exposure to blue spaces)	Cognitive and/or academic performance Restoration of attention Working memory	CPM; ANT; N-Back	(=) No associations	There were no significant relationships between childhood (school) exposure to green and/or blue spaces and any of the neurobehavioral tests used.
Largo-Wight et al. (2018) Florida, USA [53]	Quasi-experimental study N = 37 preschool students from 2 classes Children aged 3–6 years	Active exposure (outdoor classes in nature)	Restoration of attention Behavior and impulsivity Emotional well-being	Reports from children and teachers (redirections, “off-task” moments, and questionnaires)	(+) Positive	Behaviorally, there was a lower rate of redirects for classes in nature versus indoor classes (indoor classroom: 0.0834 (SD: 0.0696) vs. outdoor classroom: 0.0707 (SD: 0.0654)). In a 5 min block of classes with 20 children, the most experienced teacher observed approximately 2.5 child behavior redirections less in the outer classroom (7.5 redirections) than in the indoor classroom (10 redirections). In a block of 5 min of class with 20 children, approximately 2 fewer “off-task” incidents were observed in the outer classroom (14 off-task) compared to the indoor classroom (16 off-task) but without statistically significant differences. There were no significant differences between happiness and self-perceived well-being by the students in both conditions. However, the teachers reported that the children enjoyed their classes in nature “somewhat more” compared to the control.

Abbreviations: DECA-P2: Program *Devereux Early Childhood Assessment Preschool Program*; LOI: Learning Opportunity Index; DSM-IV: Scales of the Diagnostic and Statistical Manual of Disorders mental for to size ADHD symptoms; C-ASQ: Questionnaire abbreviated from Conner’s Symptoms for ADHD; MCAS: Massachusetts Comprehensive Assessment System.; CBCL: Behavior Checklist Child; DGS: Digit Span Memory Test; WISC: Wechsler Intelligence Scale for children; Bells Test: Letter Cancellation Test from Bell.; SCAS-P: Spence Children’s Anxiety Scale—Parent; STROOP: Stroop Test; CPT: Continuous Performance Test; SDQ: The Strengths and Difficulties Questionnaire; CASEMIS: California Special Education Management Information System; MWIA: Mental Wellbeing Impact Assessment; WCA: Wellbeing Check Cards; ANT: Attention Network Test; CICE: Composite Index of Classroom Engagement; CARS: Child Activity Rating Scale; HR: Heart Rate; d2 Test: Attention Test d2; CPM: Colored Progressive Matrix from Raven; Go/No-Go Test: Go/No-Go Reaction Test; N-Back: Working Memory Task N-Back.

## Data Availability

The data presented in this study are available on request from the corresponding author. All data used in the production of this review are included in the published studies.

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
