# Peer review of "Systematic Review: Neurodevelopmental Benefits of Active/Passive School Exposure to Green and/or Blue Spaces in Children and Adolescents"

_ijerph, 2023, doi:10.3390/ijerph20053958_

Round 1
Reviewer 1 Report
This is a nice review that really synthesizes a lot of what is published in the green/blue space literature. Overall, I think the paper could use a thorough edit as there are many grammatical errors or typos (e.g., in the abstract there is a sentence that says, "could to improve..."(remove to) and the numbers - especially when starting a sentence - should be spelled out). There is also some odd formatting that should be easily fixed (e.g., Table 1 formatting is off. The Groups info seems to be spread across two columns and it's hard to read).
In terms of content, the biggest issue is the lack of future steps in the conclusion. What does this all mean? What is the result of this work? What research or policy or interventions should take place now that we know what we know from this research? The paper would be strengthened by having that information.
Author Response
Response: Thank you very much for taking the time to review our manuscript. This new version of the manuscript was revised for grammar errors by a native English-speaking colleague and corrected appropriately, following the suggestions, see track changes. Additionally, table 1 on page 3 was formatted as suggested. Finally, regarding the future steps, we added in the discussion on page 22 the following paragraph: Despite the heterogeneity of the results, we should value the positive findings in the field of green or nature-based interventions related with the school environmental health. This work could provide diverse social and political implications for different social areas. At the policy and organizational levels, it can provide evidence to support "green" policies and guidelines related to the design and emplacement of schools in different communities, as well as practical guidelines for renaturalizing already built school environments. In addition, it could promote more sustainable management, which incorporates environmental criteria into school management contracts. At the health level, the evidence of the physical and neurodevelopmental benefits previously described, should be meant as a promotion of the school nursing’s figure. They will promote the development of activities in nature in collaboration with teachers, as well as with the school’s parents associations. In addition, the protective role of nature has special benefits when it comes from preventing pathologies associated with environmental pollution and the consumption of legal and illegal drugs, two of the most influential factors in children’s health today [64,65]. The health prescription of nature by health professionals will be indispensable in the future, creating a new way to support the health of people and nature. Promoting the concept of "One Health”. Research in this field is still scarce and uncertain; there are numerous avenues of research to deepen the benefits of nature in human health where many variables converge, leaving information gaps. It is essential to continue providing scientific evidence on how the influence of exposure to nature is related to our health, socially, physically and mentally.”.Were it explains what this all mean, what is the result of this work, what research or policy or interventions should take place now that we know from this research. Additionally, in the conclusion on page 23 we added the implications this systematic review could have.
Reviewer 2 Report
The paper addresses a crucial topic in the scientific literature regarding green/ blue spaces: the relationship between exposure to such spaces and well-being. In particular, the paper tries to highlight the nature of this relationship for what concerns children and adolescents.
The results show a general positive correlation between exposure to green/blue spaces and various aspects of well-being.
This result is unsurprising; the surprising element is that from the literature review, there are no differences between passive and active exposure.
Being a literature review, the originality of the topic, or the lack of it, is not essential to its general judgment.
The research method is thoroughly explained and scientifically sound. The literature review is solid, and the various elements considered appear well-chosen.
I suggest adding "poisonous and thorny plants" to Table 1 - "protection from pollutants and biohazards."
The main limitation of the approach is well delineated in paragraph 4.2, namely the heterogeneity of the methodologies used in the reviewed papers that limit the comparison of the outcoming data. Still, the findings seem significant for scholars and practitioners working on designing schools' outdoor spaces and those trying to improve children's and adolescents' well-being in school contexts.
The paper can be accepted in its present form.
Author Response
The paper addresses a crucial topic in the scientific literature regarding green/ blue spaces: the relationship between exposure to such spaces and well-being. In particular, the paper tries to highlight the nature of this relationship for what concerns children and adolescents.
The results show a general positive correlation between exposure to green/blue spaces and various aspects of well-being.
This result is unsurprising; the surprising element is that from the literature review, there are no differences between passive and active exposure.
Being a literature review, the originality of the topic, or the lack of it, is not essential to its general judgment.
The research method is thoroughly explained and scientifically sound. The literature review is solid, and the various elements considered appear well-chosen.
I suggest adding "poisonous and thorny plants" to Table 1 - "protection from pollutants and biohazards."
The main limitation of the approach is well delineated in paragraph 4.2, namely the heterogeneity of the methodologies used in the reviewed papers that limit the comparison of the outcoming data. Still, the findings seem significant for scholars and practitioners working on designing schools' outdoor spaces and those trying to improve children's and adolescents' well-being in school contexts.
A very meaningful and highly valued project for the health maintenance and promotion of cancer survivors.
Response: Thank you very much for taking the time to review our manuscript. This new manuscript version was checked for grammatical errors by a native English-speaking colleague and corrected accordingly, following suggestions. Additionally, in Table 1 on page 3, we added "poisonous and thorny plants" to the section "protection against contaminants and biohazards" on page 3, as suggested.
Reviewer 3 Report
This paper provides a structured review on that whether there are benefits to neurodevelopment (e.g., academic performance, attention restoration, behavior, social 614 interaction, well-being, etc.) for children’s exposure to green and blue space. In general, the paper is well-written and interesting to read. I tend to recommend this contribution after some minor revisions.
1. It might be necessary to pay attention about where each case or experiment was conducted in the included studies, since the level of human development of the country matters. In the under-developed or less urbanized regions, for example, the exposure to wild nature is more common than that in the urbanized regions. Thus the exposure may bear different functions in influencing children’s health.
2. Table 3 is unnecessarily too long by listing all references, an incorporation is possibly required, for instance, to categorize them into 7 types as you summarized in section 3.4.
3. Some discussions and specific suggestions are expected in the end of the paper regarding the implications on how these results can benefit urban design and planning in order to promote healthy and children-friendly environment in cities.
Author Response
This paper provides a structured review on that whether there are benefits to neurodevelopment (e.g., academic performance, attention restoration, behavior, social 614 interaction, well-being, etc.) for children’s exposure to green and blue space. In general, the paper is well-written and interesting to read. I tend to recommend this contribution after some minor revisions.
Response: Thank you very much for taking the time to review our manuscript. This new manuscript version was revised for grammar errors by a native English-speaking colleague and corrected appropriately, following the suggestions.
- It might be necessary to pay attention about where each case or experiment was conducted in the included studies, since the level of human development of the country matters. In the under-developed or less urbanized regions, for example, the exposure to wild nature is more common than that in the urbanized regions. Thus the exposure may bear different functions in influencing children’s health.
Response: While we appreciate the reviewer’s feedback and understand your concern. And although it is necessary to specify the results in detail, we have not differentiated the geographical regions because we lack information on both the household of the children included in the studies and the socioeconomic level of the school in general. We have reviewed articles, and not necessarily, the socioeconomic level of the neighborhood consistent with that of the children of the school. In addition, the articles do not always refer geographically to the school(s) they refer to.
- Table 3 isunnecessarily too long by listing all references, an incorporation is possibly required, for instance, to categorize them into 7 types as you summarized in section 3.4.
Response: Thank you very much for your suggestion. It is true that the table is very long, but it is the most efficient way to put the results, since if we divide by the 7 spheres of neurodevelopment that we have considered, the table would be even longer because some of the studies study several of spheres.
- Some discussions and specific suggestions are expected in the end of the paper regarding the implications on how these results can benefit urban design and planning in order to promote healthy and children-friendly environment in cities.
Response: Thank you very much for this observation. We included in the manuscript a section on the discussion and conclusion in which practical implications of the results of this study are provided from different perspectives, as suggested. See track changes.